# Effect of Bariatric Surgery on Metabolic Syndrome, Framingham Risk Scores and Thyroid Function during One-Year Follow-Up: A Saudi Retrospective Study

**DOI:** 10.3390/healthcare10122530

**Published:** 2022-12-14

**Authors:** Nuha Alamro, Afnan S. Azhri, Asma Almuqati, Firas Azzeh, Wedad Azhar, Alaa Qadhi, Najlaa H. Almohmadi, Wafaa F. Abusudah, Khloud Ghafouri

**Affiliations:** 1Clinical Nutrition Administration, King Abdullah Medical City, P.O.Box 24246, Makkah 21955, Saudi Arabia; 2Clinical Nutrition Department, Faculty of Applied Medical Sciences, Umm Al-Qura University, P.O Box 715, Makkah 21955, Saudi Arabia

**Keywords:** weight loss, cardiovascular disease, obesity

## Abstract

Bariatric surgery (BS) has been demonstrated to achieve sustained weight loss with significant metabolic improvement, including a reduction in cardiovascular disease and diabetes. The aim of this retrospective study is to measure the effect of BS on the Framingham risk score (FRS) and metabolic syndrome (MetS) among patients who underwent bariatric surgery. Additionally, we determine the effect of BS on thyroid-stimulating hormone (TSH) among euthyroid obese patients. A retrospective follow-up study was conducted at King Abdullah Medical City, Makkah, Saudi Arabia. A total of 160 patients underwent BS and completed one-year follow-up visits. Medical history, anthropometric, biochemical, and hormonal parameters were evaluated at baseline and 3–12 months after BS. The International Diabetes Federation (IDF) criteria were used to diagnose MetS. There was a significant decrease in systolic blood pressure (SBP), diastolic blood pressure (DBP), glycated hemoglobin (Hba1c), TSH, low-density lipoprotein (LDL), triglycerides, and total cholesterol (*p* < 0.001). A significant decrease was seen in MetS, BMI, FRS, SBP, DBP, Hba1c, LDL, triglycerides, cholesterol, and liver enzymes, with a significant increase in high-density lipoprotein levels 12 months postoperatively (*p* < 0.001). At 12 months, the prevalence of MetS, DM, and HTN and the FRS significantly decreased from 72.5%, 43.1%, 78.1%, and 11.4 to 16.3%, 9.4%, 22.5%, and 5.4, respectively. In addition to achieving substantial weight loss, BS improves MetS prevalence and cardiovascular risk profiles.

## 1. Introduction

Obesity is one of the most prominent epidemics of the twenty-first century; in 2016, more than 650 million were classified as obese worldwide [1]. Obesity is linked to the development of metabolic syndrome (MetS), which is s a group of conditions that together raise the risk of coronary heart disease, diabetes, stroke, and other serious health problems. MetS is characterized by elevated blood pressure, impaired glucose tolerance or insulin resistance, atherogenic dyslipidemia (i.e., high triglycerides, low high-density lipoprotein (HDL) cholesterol, high low-density lipoprotein (LDL)), and obesity [2].

MetS is strongly associated with the development of diabetes and cardiovascular disease (CVD). To prevent the occurrence of MetS among the obese population, weight loss is a key factor. Lifestyle modifications, dieting, and exercising require long-term commitment [3]. However, bariatric surgery (BS) is a rapid and the most effective approved treatment for obesity when other measures have failed [4]. BS targets either a malabsorptive or restrictive approach or a combination of both [4]. BS has been associated with metabolic changes, with the efficiency of BS to treat T2DM partially or completely having been shown to be 34–85.3% [5].

Excessive weight loss following BS has a positive impact on hypertension control. It has been observed that blood pressure is approximately 84% lower, while 51% of the BS group experienced remission of hypertension compared to 13% of the non-surgical group receiving medication therapy [6]. Favorable changes in cardiovascular risk factors have been associated with weight loss [7]. A few studies evaluated the impact of BS in reducing the 10-year Framingham risk score (FRS), which predicts the risk of a heart attack [8,9]. However, studies regarding MetS and FRS for BS patients are scarce in Saudi Arabia. 

Furthermore, thyroid hormone (TH) requirements increase with obesity, which can cause or worsen pre-existing thyroid insufficiency [10]. Obesity and plasma TSH have been found to be positively correlated in clinical studies [10,11]. Furthermore, weight loss induces a change in TH levels, particularly TSH reduction in euthyroid and hypothyroid patients [12]. Another clinical study has shown that BS can resolve hypothyroidism [13]. To the best of our knowledge, no existing data regarding TH changes following BS in euthyroid obese patients have been reported in Saudi obese subjects. Therefore, the aim of this research was to measure the effect of BS on the Framingham risk score and MetS among obese patients who underwent BS during one-year follow-up. Additionally, we determined the effect of BS on TH levels among bariatric patients.

## 2. Materials and Methods

A retrospective follow-up study was conducted among patients who underwent BS at King Abdullah Medical City (KAMC) in Makkah, Saudi Arabia, between January 2016 and December 2019. Inclusion criteria were subjects who underwent BS at KAMC and had completed 12 months of medical and nutritional follow-up visits. Additional criteria included subjects who were aged between 18 and 65 years who had undergone BS and had obesity class 1 with BMI from 30 to 35 kg/m^2^ and obesity related co-morbidities who had not achieved substantial, durable weight loss and co-morbidity improvement with reasonable nonsurgical methods or higher BMI (≥35 kg/m^2^). Patients with incomplete data records and those with major complications postoperatively were excluded. The study was approved by the KAMC Institutional Review Board (IRB number 21-757). Figure 1 displays the flowchart for the study protocol.

### 2.1. Data Collection

Data were collected from the hospital’s medical records and health information system, “Trakcare”. Demographic information and surgery type were recorded for all participants. Data included medical history, BMI, blood pressure, and laboratory results. All data were collected at baseline (preoperatively) and three months and 12 months postoperatively. The medical history of patients who were on diabetic, hypertensive, or dyslipidemia medication or thyroxine hormone replacement, or had been previously diagnosed with one of these diseases, was recorded. Laboratory results included glycated hemoglobin (Hba1c), thyroid stimulating hormone (TSH), liver enzymes (alanine aminotransferase (ALT) and aspartate aminotransferase (AST)), and lipid profile (HDL-cholesterol, LDL-cholesterol, triglycerides (TG), and total cholesterol). Fasting blood glucose was not included as the majority of the participants did not have the test postoperatively. All procedures were performed by the same group of surgeons using the same standardized techniques and instruments. For vertical sleeve gastrectomy, they used bougie calibration tube size 36 Fr. in all cases as an optimal choice to balance the effectiveness and perioperative safety of LSG. Moreover, they completely mobilized the gastric funds up to the phreno-esophageal ligament. Regarding the length of the common channel, they do not usually measure the whole length of the bowel except in revision cases; as common practice, they exclude around 100 cm as BP limb and another 100 cm as alimentary limb leaving the rest of bowel as the common channel. Risks and side effects vary by bariatric procedure. Post-op risks and side effects associated with LSG include: blood clots, gallstones (risk increases with rapid or substantial weight loss), hernia, internal bleeding or profuse bleeding of the surgical wound, leakage, perforation of stomach or intestines, stricture, vitamins, and iron deficiency. The risks of RYGB include: dumping syndrome, gallstones, perforation of stomach or intestines, pouch/anastomotic obstruction or bowel obstruction, protein or calorie malnutrition, and stomach or intestine ulceration. Our group of patients was informed and signed the surgery consent form, including the potential side effects. To reduce the procedure side effects, patients follow a very low calorie diet program at least two weeks pre-op to decrease BMI, with a strict instruction to stop smoking.

MetS was diagnosed using the five components described by the International Diabetes Federation (IDF) [14], including central obesity (defined as waist circumference for men ≥ 94 cm and for women ≥ 80 cm; if BMI is >30 kg/m^2^, central obesity can be assumed and waist circumference does not need to be measured), plus any two of the following four factors: (1) raised TG ≥ 150 mg/dL (1.7 mmol/L) or specific treatment for this lipid abnormality; (2) reduced HDL-cholesterol < 40 mg/dL (1.03 mmol/L) in males or <50 mg/dL (1.29 mmol/L) in females or specific treatment for this lipid abnormality; (3) raised systolic BP ≥ 130 or diastolic BP ≥ 85 mm Hg or treatment of previously diagnosed hypertension; and (4) raised fasting plasma glucose (FPG) ≥ 100 mg/dL (5.6 mmol/L) or previously diagnosed type 2 diabetes. A Framingham Heart Study online calculator was used to calculate the FRS to estimate the risk of developing CVD [15]. The main components for estimated FRS were age, sex, smoking status, total cholesterol, HDL, systolic BP, and use of medication for treating BP or DM.

### 2.2. Statistical Analysis

Frequency and percentages were used to present categorical variables. Minimum, maximum, and mean ± SD were used to present continuous variables. The Kolmogorov–Smirnov test was used to determine the data normality. A significant *p*-value was set below 0.05. Due to nonparametric data, *p*-values were determined by the Friedman test, with the Games-Howell post hoc test used to compare between groups. Spearman correlation was used for associations between continuous variables.

## 3. Results

A total of 160 participants (65 (40.6%) males and 95 (59.4%) females) were included in the study. The mean age was 41.9 ± 10.6 years. The mean BMI was 48.8 ± 7.3, while 27 (16.9%) patients were active smokers. There were 136 (85%) patients who had undergone laparoscopic sleeve gastrectomy, 10 (6.3%) had undergone Roux-en-Y gastric bypass, and 14 (8.8%) had undergone mini-gastric bypass. The medical history of the participants is presented in Figure 2.

Hypertension was diagnosed if systolic blood pressure was ≥140 mmHg on two different days and/or the diastolic blood pressure readings on both days were ≥90 mmHg; diabetes was diagnosed if blood glucose was ≥200 mg/dL 2 h after an Oral Glucose Tolerance Test (OGTT), and hypothyroidism was diagnosed if the TSH value exceeded 4.5 mU/L.

Figure 3 illustrates comparisons of weight and BMI preoperatively and 12 months postoperatively. A significantly lower weight (*p* < 0.001) was seen 12 months postoperatively (126.7 ± 22.5 vs. 84.5 ± 13.4). The mean percentage of weight loss was 32.6 ± 7.7. A significant decrease in BMI was also noted (*p* < 0.001) 12 months postoperatively (48.8 ± 7.3 vs. 32.7 ± 5.5). A significantly lower FRS was recorded (*p* < 0.001) 12 months postoperatively (11.4 ± 10.5 vs. 5.4 ± 4.7).

Table 1 shows the comparison of blood pressure and biomarkers related to obesity preoperatively and 3 and 12 months postoperatively. A significant drop in the mean of the following findings was observed when comparing the preoperative value with both 3-month and 12-month postoperative values: systolic blood pressure, diastolic blood pressure, and Hba1c (all *p* < 0.001). A significant decrease in AST (*p* = 0.022) and ALT (*p* = 0.012) was noted 12 months postoperatively. However, there was no significant change in TSH.

Table 2 presents the comparison of lipid profiles related to obesity preoperatively and at 3 and 12 months postoperatively. A significant drop in the mean of the following findings was observed when comparing the preoperative value with both 3-month and 12-month postoperative value: LDL, triglycerides (all *p* < 0.001), and total cholesterol (*p* < 0.002). There was no significant change in mean HDL.

Figure 4 presents a comparison of the presence of MetS components preoperatively and 12 months postoperatively. All participants were obese preoperatively compared to 119 (74.4%) who were classified as obese 12 months postoperatively. Elevated blood pressure was detected among 79 (49.4%) preoperatively compared to 36 (22.5%) 12 months postoperatively. There were 69 diabetes patients (43.1%) preoperatively compared to 15 (9.4%) 12 months postoperatively. Low HDL concentrations were present in 107 (66.9%) preoperatively compared to 84 (52.5%) 12 months postoperatively. A high level of TGs was found in 45 (28.1%) preoperatively compared to 12 (7.5%) 12 months postoperatively. McNemar’s test showed a significant decrease in the presence of MetS, which was diagnosed in 116 (72.5%) preoperatively compared to 26 (16.3%) 12 months postoperatively (*p* < 0.001 *).

The correlation between TSH and BMI showed no significant results either preoperatively (r = 0.15, *p* = 0.7) or 12 months postoperatively (r = 0.09, *p* = 0.43).

## 4. Discussion

This study aimed to measure the effect of BS on the Framingham risk score and MetS among obese patients who underwent BS. Additionally, we determined the effect of BS on thyroid hormone levels among bariatric patients. 

The prevalence of MetS at baseline was 72.5%. Our results were higher compared to studies conducted locally, with 39.8% [16], and internationally, including 66.5% in China [17] and 54% in Egypt [18]. The higher prevalence in our study might relate to the fact that all our participants were obese. All these studies demonstrated a positive association between obesity and MetS. MetS resolution was achieved in 72.5% of patients one year postoperatively, which agrees with a previous study that found that 78% of patients who underwent BS recovered from MetS [19]. Similarly, a previous study reported remission in MetS by 75.8% of patients one year after RYGB [20].

A significant drop in mean BMI, by 33%, was seen in the current study. This is consistent with previous studies [21,22]. A significant reduction in FRS was reported postoperatively by 46% of patients compared to 40% in previous studies [23,24].

Hypertension is a MetS component and is linked to obesity [25]. Many studies have demonstrated that weight loss lowers blood pressure [26,27]. One year after BS, 71.2% of hypertensive patients had recovered and discontinued treatment. Furthermore, systolic blood pressure had decreased by 6% three months postoperatively and by 10% 12 months postoperatively. Diastolic blood pressure had dropped significantly by 4% and 5% at 3 months and 12 months operatively, respectively. A large meta-analysis showed improvement in hypertension in 63.7% of patients. Additionally, hypertension resolution was reported in 50% of patients [28]. Similarly, it has been found that LSG can improve and resolve elevated blood pressure in the short and long term [29].

BS offers a safe and effective treatment in achieving sustained glycemic control in diabetic obese patients, while decreasing chronic micro- and macroangiopathic complications [30,31]. In the current study, complete resolution of T2DM was achieved in 78.2% of diabetic patients. A meta-analysis confirmed that BS led to 72% remission [32]. Three important anatomical changes after BS may initiate T2DM remission. Gastric restriction is the first anatomic change, inducing a decrease in caloric intake that can improve hepatic insulin sensitivity and reduce hepatic gluconeogenesis, which may contribute to the removal of glucotoxic effects on pancreatic beta cells, as illustrated by the rapid enhancement of insulin secretion in patients with T2DM [33]. After LSG, ghrelin reduction may be important in the prevention of weight regain, with ghrelin shown to have diabetogenic effects by suppressing insulin secretion and possibly inducing hyperglycemia. However, ghrelin is undoubtedly decreased long term after fundus resection, which may play an important role in the sustainability of weight loss after LSG [34]. Resection of the duodenum and upper intestine is the second anatomical change that may reduce fat absorption and change the entero-hepatic flow of bile acid. Rapid transport of food to the distal bowel is the third anatomic change that will induce glucagon-like peptide-1 (GLP-1) and peptide YY (PYY) changes but, more importantly, may alter bile acid recycling and increase blood bile acid concentrations [35]. Further studies involving the application of phenotyping, genomics, metabolomics, and gut microbiome studies will enhance our understanding of metabolic surgeries and help identify novel therapeutic targets. 

Before surgery, 15.6% [25] of patients had dyslipidemia. Total cholesterol, LDL-cholesterol, and triglyceride levels were significantly lower at 3 and 12 months postoperatively than preoperatively. A non-significant increase in HDL concentration was seen 12 months postoperatively. In agreement with the present results, a significant improvement was found in total cholesterol, LDL-cholesterol, and triglyceride levels (at 3, 6, 9, 12, 18, and 24 months) compared to preoperative. They also reported that HDL-cholesterol increased significantly from 6 months onward, reaching its highest value 24 months postoperative [36]. The improvement in the lipid profile might be due to the malabsorptive influence of BS [37]. In malabsorptive surgery, cholesterol absorption decreases while cholesterol synthesis increases, which is associated with enhanced hepatic catabolism [38]. Fat malabsorption leading to increased fecal fat excretion and elevated intestinal oxalate absorption induced by RYGB have been implicated in the improvement of postprandial lipemia in patients undergoing this type of surgery [38]. For restrictive procedures, a reduction in gastric volume and a decrease in the production of gastric lipase and cholecystokinin result in decreased hydrolysis of triglycerides, leading to reduced absorption of free fatty acids [38]. 

The current study found a significant improvement at 12 months postoperatively in both AST and ALT serum levels. This agrees with earlier studies [39,40,41]. The underlying mechanisms by which liver improvement occurs need to be further investigated.

Several studies have investigated the relationship between obesity and TSH levels in euthyroid obese patients [42,43,44]. Normal values range between 0.5–5.0 mIU/L. In our study, no significant improvement in TSH levels was noticed postoperatively. This is not in agreement with other results reported previously [45]. Neves et al. found a significant decrease in TSH levels 12 months postoperatively in the high–normal TSH group and normal TSH group [46]. Studying the seasonal variation of vitamin D and thyrotropin levels and its relationship in a euthyroid Caucasian population, Das et al. found that TSH values follow a seasonal pattern, decreasing during the summer and then increasing during the winter [47]. These annual variations in TSH secretion should be investigated in the Saudi Arabian population.

No significant correlation was found between BMI and TSH level preoperatively or 12 months postoperatively. This finding is similar to previous findings [48,49]. The mechanisms of TSH decrease after BS remain unclear. The decrease in TSH is probably mediated by weight loss and is not due to an intrinsic effect of BS. A decrease in TSH levels was found in obese patients after exercise, behavior therapy, and calorie-restricted-induced weight loss [50]. This study shows that in euthyroid morbidly obese patients, the acute phase after BS promotes a significant decline in moderately increased TSH levels. More studies should be conducted to understand the underlying mechanism of TSH and obesity. This is a retrospective follow-up study examining a variety of metabolic outcomes following BS. The data are representative of the specialized surgical center for BS in KAMC and reflect current medical practice. Therefore, the agreement of our results with other local and international studies is likely to be generalizable to the Makkah region population. Considering the type of surgical technique is beyond the scope of this study. In recent years, several authors have reported excellent short-term results with performing sleeve gastrectomy, but some aspects regarding the variability of gastric tubulization design could influence the results obtained in relation to weight loss and functional changes and gastric hormones. A systematic review and network meta-analysis were conducted to assess the weight loss effects and associated complications of LSG for patients with morbid obesity; based on different bougie sizes, they found that using M-sized bougie (33–36 Fr.) is an optimal choice to balance the effectiveness and perioperative safety of LSG in the clinical practice, which was used with all our participants [51].

There are some limitations to our study, including that we cannot control exposures such as caloric intake, dietary behavior, or physical activity. Due to incomplete data and the retrospective nature of our project, the correlation between TSH level and percentage of weight loss was not studied, as thyroid tests were not available for all euthyroid patients after surgery. Because of the study nature, the information about the control group without surgery is limited. Of our euthyroid patients, 80% had a TSH serum test 3 months postoperatively, while 54% had a TSH serum test 12 months postoperatively, compared to 98% preoperatively.

Another potential limitation in the interpretation of these results is the inequality in the type of BS; for example, LSG was a higher producer to preformed. Previous analyses have shown that RYGB results in superior weight loss compared to LSG. Additionally, another limitation in the statistical analysis was that no correction was made for multiple comparisons. 

## 5. Conclusions

To conclude, this study contributes to the understanding of the metabolic changes after BS and profoundly elucidates the effect of obesity on MetS. The findings agree that BS is a powerful option in the treatment of obesity and associated complications. Further randomized controlled trials are needed to investigate the short- and long-term effects of BS on weight maintenance and the resolution of comorbidities as a function of the type of BS (malabsorptive and restrictive procedures).

## Figures and Tables

**Figure 1 healthcare-10-02530-f001:**
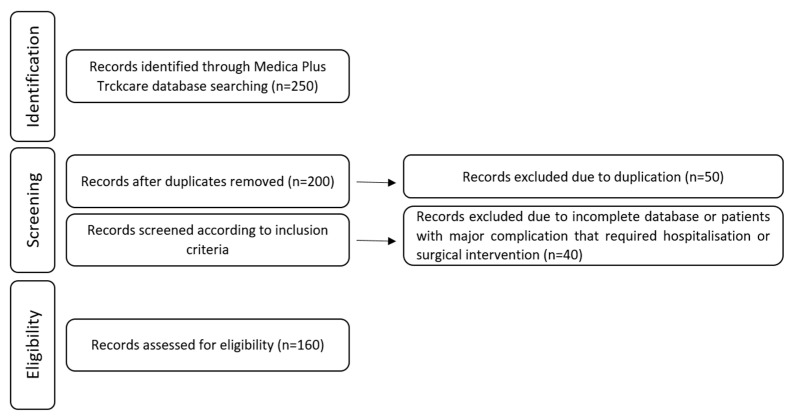
Flowchart of study protocol.

**Figure 2 healthcare-10-02530-f002:**
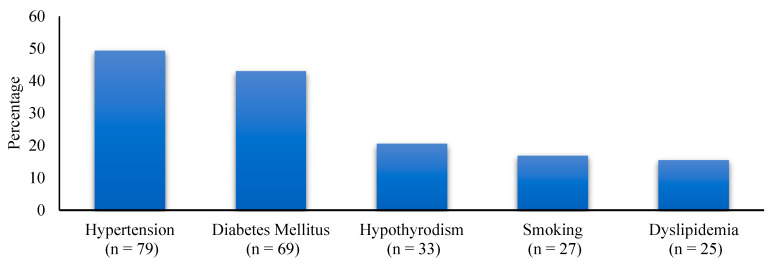
Medical history of the participants.

**Figure 3 healthcare-10-02530-f003:**
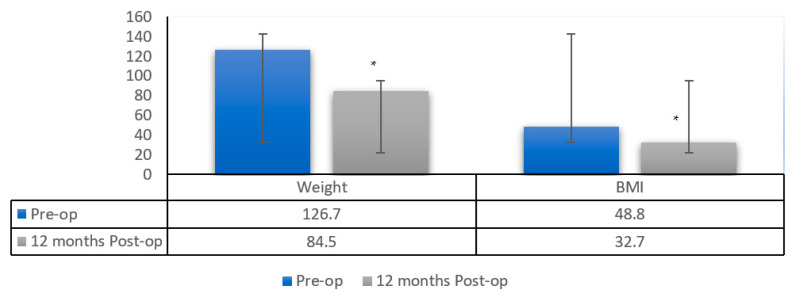
Comparison of weight and BMI pre-op and 12 months post-op. * Significant difference when compared with the preoperative value at *p* < 0.05.

**Figure 4 healthcare-10-02530-f004:**
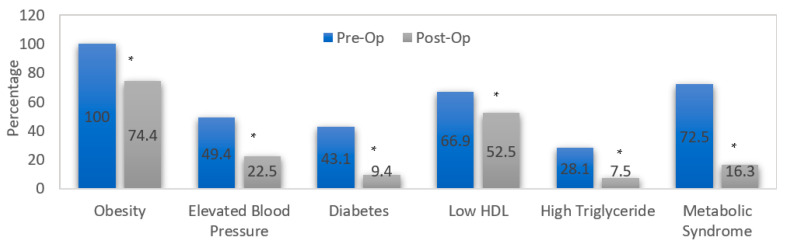
Comparison in the presence of MetS components pre-op and 12 months post-op. * Significant difference when comparing with the preoperative value at *p* ≤ 0.001.

**Table 1 healthcare-10-02530-t001:** Comparison of blood pressure and biomarkers related to obesity across a 12-month period postoperatively (n = 160).

Parameter	Preoperative	3 Months	12 Months	*p*-Value
**Systolic Blood Pressure (mm Hg)**	139.5 ^a^ ± 13.7	130 ^b^ ± 9.9	125.1 ^c^ ± 7.6	<0.001 *
**Diastolic Blood Pressure (mm Hg)**	75.4 ^a^ ± 9.7	71.8 ^b^ ± 8.7	71.6 ^b^ ± 7.6	<0.001 *
**Hba1c** **(%)**	6.9 ^a^ ± 1.8	6 ^b^ ± 1.4	5.4 ^c^ ± 0.8	<0.001 *
**TSH (mIU/L)**	2.8 ± 2.4	2.3 ± 1.4	3.0 ± 1.6	0.388
**AST (units/L)**	23.6 ^a^ ± 16	23.2 ^a^ ± 15.4	19.6 ^b^ ± 11.8	0.022 *
**ALT (IU/L)**	31.6 ^a^ ± 23.5	31.7 ^a^ ± 28.1	24.9 ^b^ ± 15.8	0.012 *

* *p*-values were determined by Friedman test. Means in the same row with different superscripts are significantly (*p* < 0.05) different according to Games-Howell post hoc test.

**Table 2 healthcare-10-02530-t002:** Comparison of lipid profiles across a 12-month period postoperatively (n = 160).

Parameter	Preoperative	3 Months	12 Months	*p*-Value
**HDL (mg/dL)**	56 ± 14.8	56.6 ± 18	59 ± 13.4	0.199
**LDL (mg/dL)**	126 ^a^ ± 49.4	112.9 ^b^ ± 39.5	96.9 ^c^ ± 28.9	<0.001 *
**Triglycerides (mg/dL)**	126.6 ^a^ ± 53.9	107.7 ^b^ ± 40.4	92.4 ^c^ ± 36.4	<0.001 *
**Total Cholesterol (mg/dL)**	205.1 ^a^ ± 52.8	189.7 ^b^ ± 40.9	176.3 ^b^ ± 106.8	0.002 *

* *p*-values were determined by Friedman test. Means in the same row with different superscripts are significantly (*p* < 0.05) different according to Games-Howell post hoc test.

## Data Availability

Not applicable.

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
