# Peer review of "Effect of Bariatric Surgery on Metabolic Syndrome, Framingham Risk Scores and Thyroid Function during One-Year Follow-Up: A Saudi Retrospective Study"

_healthcare, 2022, doi:10.3390/healthcare10122530_

Round 1

Reviewer 1 Report

Methods: line 68. Is BS performed on patients with Class 1 obesity?

Results: Changes over Three time line can be compared using repeated measure ANOVA or Friedman  test for non parametric data. Perform post hoc test to identify the difference between time lines.

To the study the association between  BMI and TSH did the authors also consider correlating percentage of weight lost with TSH?

Indicate significant difference in Figure 3.

Please refer to the attached file for further comments. 

Author Response

Reviewer #1

  1. Methods: line 68. Is BS performed on patients with Class 1 obesity?

The answer:

As mentioned in line 70, The American Society of Metabolic and Bariatric Surgery updated their position on qualifiers for weight loss surgery to include Class I obese patients. BMI from 30 to 35 kg/m2 and obesity related co-morbidities who do not achieve substantial, durable weight loss and co-morbidity improvement with reasonable nonsurgical methods.

  1. Results: Changes over Three time line can be compared using repeated measure ANOVA or Friedman test for non parametric data. Perform post hoc test to identify the difference between time lines.

The answer:

The statistical analysis section is updated as seen in the corrected manuscript (lines 109-115).

  1. To the study the association between BMI and TSH did the authors also consider correlating percentage of weight lost with TSH?

The answer:

As mentioned in 267, as one of the limitation, the correlation between TSH level and percentage of weight loss weren’t studied as thyroid tests were not available for all euthyroid patients after surgery.

  1. Indicate significant difference in Figure 3.

The answer:

Done in the corrected manuscript

  1. Please refer to the attached file for further comments. 

The answer:

Done in the corrected manuscript

Finally, we want to thank you for your valuable comments, and hope all corrections are performed correctly.

Sincerely Yours,

The research team

November/26/2022

Reviewer 2 Report

Alamro and colleagues conduct a retrospective study to investigate the relationship between bariatric surgery and metabolic syndrome. It is important topic, but the information may need some improvement to provide more solid result to the readers. Some suggestions were listed below. 

1. Is it possible to provide the information of the control group without surgical intervention.

2. Is all the procedure provided by the same group of surgeons or standardized steps in your center? There may be some factors affecting the effect of bariatric metabolic surgery, such as the size of bougie calibration(which was discussed recently in an review article in Scientific Report) , the fundus exclusion, the length of common limbs, etc.

3. Is there any adverse events related to the bariatric procedure? I think it is also important to discuss the risk of surgical intervention and to prove the safety of bariatric surgery. 

4. It would improve the readability to provide a flowchart of study protocol, such as STROBE flowchart.

Author Response

Reviewer #2

  1. Is it possible to provide the information of the control group without surgical intervention.

The answer:

It’s can’t be possible due to the nature of the study “”retrospective-follow up” , and this valuable point is added to the limitation section.  

  1. Is all the procedure provided by the same group of surgeons or standardized steps in your center? There may be some factors affecting the effect of bariatric metabolic surgery, such as the size of bougie calibration(which was discussed recently in an review article in Scientific Report) , the fundus exclusion, the length of common limbs, etc.

The answer:

As mentioned in line 87, All procedure were done by the same group of surgeons using the same standardized techniques and instruments, for vertical sleeve gastrectomy they are using bougie calibration tube size 36 Fr. in all cases as an optimal choice to balance the effectiveness and perioperative safety of LSG. Also they are completely mobilizing the gastric funds up to the phreno-esophageal ligament. Regarding to the length of common channel they are not usually measuring the whole length of the bowel except in revision cases, as common practice they exclude around 100 cm as BP limb and another 100 cm as Alimentary limb leaving the rest of bowel as common channel.    

  1. Is there any adverse events related to the bariatric procedure? I think it is also important to discuss the risk of surgical intervention and to prove the safety of bariatric surgery. 

The answer:

Bariatric surgery has substantial and sustained effects on weight and significantly ameliorates obesity-attributable comorbidities in the majority of bariatric surgery patients. However, complication rates associated with bariatric surgery range from 10% to 17% and reoperation rates approximately 7%; nonetheless, mortality associated with surgery is generally low (0.08%-0.35%). According to updated systemic review and meta-analysis published in 2014

  1. It would improve the readability to provide a flowchart of study protocol, such as STROBE flowchart.

The answer:

Thank you for your valuable point. The study protocol is fully considered in the methodology section, and we see that the flowchart will not represent the study protocol clearly because of its limitation by the retrospective nature of this study.

Finally, we want to thank you for your valuable comments, and hope all corrections are performed correctly.

Sincerely Yours,

The research team

November/26/2022

Reviewer 3 Report

The work is correctly written. The idea on which it is based may not be novel, but it reliably provides further evidence of the beneficial effects of BS in reducing cardiovascular risk as assessed objectively by the Framingham Scale. As the authors emphasize, this is the first work of this type on the population of Saudi Arabia, so its results may be helpful to clinicians in this area. A few editorial errors and typos (e.g. in Figure3, obesity in the word, in the text line 246 is THS instead of TSH).

Author Response

Reviewer #3

The work is correctly written. The idea on which it is based may not be novel, but it reliably provides further evidence of the beneficial effects of BS in reducing cardiovascular risk as assessed objectively by the Framingham Scale. As the authors emphasize, this is the first work of this type on the population of Saudi Arabia, so its results may be helpful to clinicians in this area. A few editorial errors and typos (e.g. in Figure3, obesity in the word, in the text line 246 is THS instead of TSH).

The answer:

All editorial errors and typos are changed in the corrected manuscript.

Finally, we want to thank you for your valuable comments, and hope all corrections are performed correctly.

Sincerely Yours,

The research team

November/26/2022

Round 2

Reviewer 2 Report

I am really grateful for the authors valuable work to improve the manuscript. However, I believed the manuscript could be better, and even the manuscript had a retrospective nature. Such as, the study protocol could be visualized with a flowchart, which could improve the readability in my opinion. I also believed that it is important to report the related adverse events of the surgical procedure, not merely the data from meta-analysis.( The data from every surgical team is valuable and important for the readeres.) Also, I thought the detail about the surgical procedure could be discuss in the discussion section because some details could make contribution to the weight loss effect, such as fundus exclusion, smaller bougie size, gut hormone, etc. 

Author Response

Dear respected Editor

 Healthcare Journal

Thank you for reviewing our paper entitled “Effect of Bariatric Surgery on Metabolic Syndrome, Framingham Risk Scores and Thyroid Function during One-year Follow-up: A Saudi Retrospective Study”. We have reviewed and corrected our paper based on your notes.

All reviewer comments in the revised manuscript have been changed.

Concerning your comments, the following is our reply to reviewers:

1) the study protocol could be visualized with a flowchart, which could improve the readability in my opinion.

The answer

Added lines 75 and 120-130 in the corrected manuscript.

2) I also believed that it is important to report the related adverse events of the surgical procedure, not merely the data from meta-analysis. (The data from every surgical team is valuable and important for the readeres)

The answer

Added lines 95-105 in the corrected manuscript.

3) Also, I thought the detail about the surgical procedure could be discuss in the discussion section because some details could make contribution to the weight loss effect, such as fundus exclusion, smaller bougie size, gut hormone, etc

The answer

Added lines 245-258 as well as 303-310 in the corrected manuscript.

Finally, we want to thank you for your valuable comments, and hope all corrections are performed correctly.

Sincerely Yours,

The research team

December/8/2022